# Secondary analyses to test the impact on inequalities and uptake of the schools-based human papillomavirus (HPV) vaccination programme by stage of implementation of a new consent policy in the south-west of England

Harriet Fisher ![ORCID],[1] Karen Evans,[2] Rosy Reynolds,[1] Julie Yates,[3] Marion Roderick,[4] Jo Ferrie,[3] John Macleod,[1,5] Matthew Hickman ![ORCID],[1] Suzanne Audrey[1]

► Prepublication history and additional online supplemental material for this paper are available online. To view these files, please visit the journal online (http://dx.doi.org/10.1136/bmjopen-2020-044980).

For numbered affiliations see end of article.

**Correspondence to**
Dr Harriet Fisher;
Harriet.Fisher@bristol.ac.uk

## ABSTRACT

**Objectives** To test the impact on inequalities and uptake of the schools-based human papillomavirus (HPV) vaccination programme by stage of implementation of a new policy providing additional opportunities to consent.

**Setting** Two local authorities in the south-west of England.

**Participants** Young women (n=7129) routinely eligible for HPV vaccination aged 12–13 years during the intervention period (2017/2018 to 2018/2019 programme years).

**Interventions** Local policy change that included additional opportunities to provide consent (parental verbal consent and adolescent self-consent).

**Outcomes** Secondary analyses of cross-sectional intervention data were undertaken to examine uptake by: (1) receipt of parental written consent forms and; (2) percentage of unvaccinated young women by stage of implementation.

**Results** During the intervention period, 6341 (89.0%) eligible young women initiated the HPV vaccination series. Parental written consent forms were less likely to be returned where young women attended alternative education provider settings (p<0.001), belonged to non-white British ethnic groups (p<0.01) or more deprived quintiles (p<0.001). Implementation of parental verbal consent and adolescent self-consent reduced the percentage of unvaccinated young women from 21.3% to 16.5% (risk difference: 4.8%). The effect was greater for young women belonging to the most deprived compared with the least deprived quintile (risk difference: 7.4% vs 2.3%, p<0.001), and for young women classified as Unknown ethnic category compared with white British young women (6.7% vs 4.2%, p<0.001). No difference was found for non-white British young women (5.4%, p<0.21).

**Conclusions** Local policy change to consent procedures that allowed parents to consent verbally and adolescents to self-consent overcame some of the barriers to vaccination of young women belonging to families less likely to respond to paper-based methods of gaining consent and at greater risk of developing cervical cancer.

### Strengths and limitations of this study

► This study uses routinely collected data to examine the impact of local policy changes on inequalities in uptake of the human papillomavirus vaccination programme.

► The intervention period (programme years 2017/2018–2018/2019) was relatively short.

► Missing data on ethnicity could change the direction or size of the corresponding ORs.

**Trial registration number** 49 086 105.

## INTRODUCTION

The human papillomavirus (HPV) vaccine currently used in England protects against infection from high-risk HPV types 16 and 18 which cause cancers affecting the cervix, vulva, vagina, penis, anus and oral cavity. The vaccine also protects against types 6 and 11 which cause 90% of genital warts. High coverage of the English HPV vaccination programme for young women aged 12–13 years has been achieved. Recent evidence highlights the potential for HPV vaccination programmes to substantially reduce the incidence of cervical cancer.[1 2] Based on emerging evidence for cost-effectiveness, in 2019/2020 the HPV vaccination programme was expanded to include young men aged 12–13 years.

Despite generally good coverage, without concerted efforts to address lower uptake among some populations, pre-existing disparities in the incidence of cervical cancer by ethnicity and deprivation may increase.[3–5] We previously identified lower uptake by area

and among some population groups, including minority ethnic groups.[6] Our research in schools with lower uptake showed complex socio-cultural factors can influence whether young women are vaccinated[7] and the requirement for written parental consent may act as a barrier to some young women receiving the HPV vaccine.[8]

In the UK (comprising England, Scotland, Northern Ireland and Wales), consent for schools-based adolescent vaccination programmes is usually obtained from parents or carers. Young people are provided with a form for their parent to sign and hand back to school before a vaccination session takes place. However, the UK legal framework allows young people to be vaccinated without parental consent provided they are deemed 'Gillick competent'[9] (ie. they have sufficient maturity and intelligence to understand the nature and implications of the treatment). Findings from an evidence synthesis showed the implementation of adolescent self-consent procedures could be prevented by local policies which favoured written parental consent, and the precedence given to professionals' concerns about their reputations and relationships with parents.[10]

A local authority is an organisation officially responsible for all public services and facilities in a particular area. To address concerns about lower uptake in two local authority areas in the south-west of England, a new policy (including parental verbal consent and adolescent self-consent) has been implemented.[9] Implementation of the new consent procedures in one of the intervention areas appeared to improve uptake in contrast to trends of decreasing uptake among matched local authorities.[11] However, no evidence for an absolute increase, or reduction in inequalities by deprivation and ethnicity was found.

The aim of the current study is to further examine the impact of the consent procedures on HPV vaccine uptake. Specifically, we describe:
1. Receipt of parental written consent forms by school category, ethnicity and deprivation quintile.
2. Unvaccinated young women by stage of implementation of consent procedures, and by school category, ethnicity and deprivation quintile.

## METHODS
This cross-sectional study was undertaken when the English vaccination programme was delivered routinely to young women only. Details of the evaluation and changes to the new local policy are provided in a published protocol.[12]

### The new local policy (including consent procedures)
In brief, previously only young women who had returned a written parental consent form, indicating the parent or carer is willing for their daughter to receive the HPV vaccine, were administered the vaccine in the school setting. Under the new arrangements, all young women eligible for the vaccination, including those whose parents have provided written refusal, are asked to attend the session by the immunisation team. For those young women who do not have a returned parental written consent form, the immunisation nurse attempts to gain parental verbal consent over the telephone. If the parent cannot be contacted and the young woman expresses willingness to be vaccinated, the immunisation nurse assesses the young woman's competence and if they are deemed competent the young women will receive the vaccine. All young women who do not receive the vaccine on the day are given information about alternative options to receive the vaccination, such as through their family doctor or community-based clinics run by the immunisation team (online supplemental material 1). Parents are not routinely contacted again by the immunisation team.

We define stage of implementation of consent procedures as the following: (1) 'stage 1: parent written consent only'; (2) 'stage 2: parent verbal consent and adolescent self-consent' and; (3) 'stage 3: community catch up clinics and family practice settings'. These stages represent sequential opportunities (in a single programme) for unvaccinated young women to receive the vaccine, rather than different time points.

### Patient and public involvement
The Bristol Young Person's Advisory Group (YPAG) (website: https://generationr.org.uk/bristol/) took part in preliminary discussions about self-consent procedures, at which a group of 11 young people discussed self-consent for vaccination of those of secondary school age. Following this discussion, all voted in favour of self-consent preferably with parents being informed. The young people were overall positive about the documents which they thought provided the right amount of information clearly. As a result of the feedback, some changes to the wording of the information and formatting were made. Members of the Bristol Youth Council (website: https://www.bristol.gov.uk/youth-council-youth-mayors) were approached to participate in the study during recruitment. This resulted in an opportunity to obtain feedback on a proposal developed from the findings of the current study.

During the study, the lead researcher was also invited to deliver school-based information sessions about the HPV vaccine to year 7 students. The study researcher also led PSHE (Personal, Health, Social and Economic) lessons where year 8 students were able to find out about research and encouraged to debate issues around adolescent self-consent. Study findings are being shared with the YPAG and the Bristol City Youth Council at meetings to mark the end of the study.

### Population
Two local authorities implementing the new consent procedures for the HPV vaccination programme in the south-west of England provided data. Records relating to young women eligible (born between 1 September 2004 and 31 August 2006) for vaccination during programme years 2017/2018 and 2018/2019 and who were registered

with a general practice within the local authority boundaries were retrieved in July 2019 from the Child Health Information System.

## Data extraction from the Child Health Information System

Prior to study commencement, permission to access an anonymised data extract was gained from the relevant organisations with responsibility for the data. In the UK, the Child Health Information System holds demographic and vaccination-related records for each young person registered with a family doctor which is a statutory requirement. The following data fields were extracted from records of the eligible population: (1) partial date of birth; (2) partial postcode; (3) ethnicity; (4) dates and location HPV vaccination administered and (5) name and corresponding identifying code of school.

School identifying codes were used to assign the local authority responsible for delivery of the HPV vaccine. Partial date of birth was used to allocate programme year the young woman was eligible to receive the HPV vaccine. Categories of school types were applied to each record: (1) comprehensive, non-fee paying; (2) private, fee paying and (3) alternative education provider, which included pupil referral units, young offender units, hospital education service, specialist schools for students with significant additional needs and young women educated at home.

Individual records were classed as 'received HPV vaccine' if there was a record of at least one dose administered within the corresponding programme year the young woman was eligible. Postcodes from individual records were linked to the corresponding lower super output area. Deprivation score was assigned using the Index of Multiple Deprivation 2019 (a statistic on relative deprivation in small areas of England)[13] and analysed as quintiles. Due to small numbers, ethnicity was grouped as follows: (1) white British; (2) non-white British and (3) unknown.

Records were excluded if the relevant school identifying code was missing or invalid. Absence of recorded ethnicity was considered likely Missing Not At Random as absence of ethnicity data was associated with the outcome, school and deprivation variables. A complete case approach where records were excluded on the basis of missing ethnicity is not recommended.[14] Instead, the 'unknown' category was assigned to missing ethnicity data to minimise the risk of bias.

## Data extraction from the immunisation team's records of consent

Additional data were sought from the school immunisation team's electronic and paper-based records relating to vaccination consent during the intervention period. This included: (1) return of parental consent forms ('yes' or 'no') and; (2) stage of implementation of consent procedures. We classified each record as belonging to one of the following stages of the consent procedure: (1) 'stage 1: parent written consent only'; (2) 'stage 2: parent verbal consent and adolescent self-consent' and (3) 'stage 3:

community catch up clinics and family practice settings'. Records could not be assigned to more than one stage of consent category.

## Data linkage

The data extracted from the immunisation team's records were linked to the Child Health Information System using deterministic data linkage methods by a member of staff at Health Intelligence. An anonymised version of the data extract was securely transferred to researchers at the University of Bristol.

## ANALYSIS
### Return of parental consent forms

Logistic univariable analyses and likelihood ratio tests were performed to explore factors associated with return of parental consent form. The following explanatory variables for analysis were selected a priori: school category, ethnicity and deprivation quintile. A multivariable logistic regression model was developed. We used cluster-robust errors in the final model to allow for the possibility of clustering within schools.

### Unvaccinated young women by stage of implementation of consent procedures

To describe the decrease in unvaccinated young women at each stage of implementation of the consent procedures, we calculated risk differences (difference in two proportions) with 95% Confidence Intervals. The risk difference shows the absolute effect of implementation of each stage of the consent procedure. We considered the following risk differences (risk reductions) by: (1) percentage of young women unvaccinated during 'stage 1: parent written consent only' minus percentage of young women unvaccinated during 'stage 1' and 'stage 2: parent verbal consent and adolescent self-consent' and; (2) percentage of young women unvaccinated during 'stage 1' minus percentage of young women unvaccinated during 'stage 1', 'stage 2' and 'stage 3: community catch up clinics and family practices').

To show whether there was an unintended increase or reduction in health inequalities, we compared the risk differences and corresponding p values by school category, ethnic group and deprivation quintile—comparing with a baseline category in each case.

Analyses were undertaken using the Stata, release V.15 (Stata).

## RESULTS

Data were extracted relating to 7549 young women eligible for vaccination during the intervention period (programme years 2017/2018–2018/2019). Of these, 420 (5.6%) were excluded on the basis that the school data was missing or invalid.

Of the cohort retained for analysis (n=7129), the majority of vaccine eligible young women were resident in

**Table 1** Descriptive summary of eligible cohort by vaccine receipt and return of parent consent form

| | Eligible cohort | HPV vaccine received | Unreturned parent consent form |
|---|---|---|---|
| | n (%) | n (%) | n (%) |
| | 7129 (100.0) | 6341 (89.0) | 1155 (16.2) |
| **Area level** | | | |
| Local authority 1 | 4516 (63.4) | 3944 (87.3) | 843 (18.7) |
| Local authority 2 | 2613 (36.7) | 2397 (91.7) | 312 (11.9) |
| Programme year 2017/2018 | 3581 (50.2) | 3202 (89.4) | 565 (15.8) |
| Programme year 2018/2019 | 3548 (49.8) | 3139 (88.5) | 590 (16.6) |
| **School category** | | | |
| Comprehensive, non-fee paying | 6350 (89.1) | 5690 (89.6) | 992 (15.6) |
| Private, fee paying | 661 (9.3) | 565 (85.5) | 105 (15.9) |
| Alternative education providers | 118 (1.7) | 86 (72.9) | 58 (49.2) |
| **Individual level** | | | |
| Ethnicity | | | |
| White British | 4888 (68.6) | 4552 (93.1) | 610 (12.5) |
| Non-white British | 572 (8.0) | 492 (86.0) | 101 (17.7) |
| Unknown | 1669 (23.4) | 1297 (77.7) | 444 (26.6) |
| Deprivation | | | |
| Least deprived | 1348 (18.9) | 1235 (91.6) | 136 (10.1) |
| Quintile 2 | 1379 (19.3) | 1277 (92.6) | 149 (10.8) |
| Quintile 3 | 1403 (19.7) | 1273 (90.7) | 210 (15.0) |
| Quintile 4 | 1396 (19.6) | 1203 (86.2) | 292 (20.9) |
| Most deprived | 1421 (19.9) | 1194 (84.0) | 338 (23.8) |
| Unknown | 182 (3.6) | 159 (87.4) | 30 (16.5) |

HPV, human papillomavirus.

local authority one (4516, 63.4%), attended comprehensive, non-fee paying schools (6350, 89.1%), and were classified as belonging to a white British ethnic group (4888, 68.6%). Of young women eligible for vaccination, 6341 (89.0%) were recorded to have received the HPV vaccine during the programme year they were eligible. Parental consent forms were recorded as being unreturned (comprising active refusal and passive non-consent) for 1555 (16.2%) of eligible young women (table 1).

### Return of parental consent forms
Variables associated with return of parental consent forms are provided as unadjusted ORs in table 2. After adjusting for school category, ethnicity and deprivation, an association was found between parental consent form not being returned and: attending an alternative education provider setting (adjusted OR (aOR): 5.54, 95% CI: 3.80 to 8.09, p<0.001), belonging to a non-white British (aOR: 1.34, 95% CI: 1.06 to 1.70, p<0.01) and unknown ethnicity category (aOR: 2.41, 95% CI: 2.09 to 2.78, p<0.001). There was also evidence for a relationship with level of deprivation. For example, young women belonging to the most deprived quintile had at least double the odds of having a record of unreturned consent form (aOR: 2.54,

95% CI: 2.03 to 3.18, p<0.001) compared with those from the least deprived quintile (table 2).

### Percentage of young women unvaccinated by stage of implementation of consent procedures
Following implementation of stage 1 (parental written consent) of the consent procedures, 1519 (21.3%) of young women were unvaccinated. At stage 2 (parental verbal consent and adolescent self-consent), this reduced to 1173 (16.5%) unvaccinated young women. With the inclusion of stage 3 (community catch up clinics and family practice settings), there remained 788 (11.1%) unvaccinated young women during the study period (table 3). Not all parents could be contacted by the immunisation team on the day of the vaccination session (n=362). These parents were not routinely contacted again by the immunisation team (data not shown).

The percentage of unvaccinated young women varied by school category at different stages of implementation of the consent procedures. For example, at stage 1 (parental written consent only), 20.6% of young women who attended mainstream comprehensive, non-fee-paying schools were unvaccinated, in comparison to 50.8% who attended alternative education provider settings. At stage

**Table 2** Associations of unreturned parental consent form with school category, ethnicity and deprivation

| | N (%) | Form not returned | OR (95% CI)* | P value | aOR (95% CI)* | P value |
|---|---|---|---|---|---|---|
| | | | – | – | – | – |
| **School category** | | n (%) | | | | |
| Comprehensive, non-fee paying | 6350 | 992 (15.6) | – | | – | |
| Private, fee paying | 661 | 105 (15.9) | 1.02 (0.82 to 1.27) | 0.86 | 1.12 (0.89 to 1.43) | 0.34 |
| Alternative education providers | 118 | 58 (49.2) | 5.22 (3.62 to 7.54) | <0.001 | 5.54 (3.80 to 8.09) | <0.001 |
| **Ethnicity** | | | | | | |
| White British | 4888 | 610 (12.5) | – | – | – | – |
| Non-white British | 572 | 101 (17.7) | 1.50 (1.19 to 1.89) | <0.01 | 1.34 (1.06 to 1.70) | 0.01 |
| Unknown | 1669 | 444 (26.6) | 2.54 (2.21 to 2.92) | <0.001 | 2.41 (2.09 to 2.78) | <0.001 |
| **Deprivation** | | | | | | |
| Least deprived | 1279 | 136 (10.1) | – | | – | |
| Quintile 2 | 1324 | 149 (10.8) | 1.08 (0.84 to 1.38) | 0.54 | 1.09 (0.85 to 1.40) | 0.48 |
| Quintile 3 | 1334 | 210 (15.0) | 1.57 (1.25 to 1.97) | <0.001 | 1.57 (1.24 to 1.98) | <0.001 |
| Quintile 4 | 1305 | 292 (20.9) | 2.36 (1.89 to 2.93) | <0.001 | 2.24 (1.79 to 2.81) | <0.001 |
| Most deprived | 1347 | 338 (23.8) | 2.78 (2.24 to 3.45) | <0.001 | 2.54 (2.03 to 3.18) | <0.001 |
| Unknown | 182 | 30 (16.4) | 1.75 (1.14 to 2.70) | 0.01 | 1.52 (0.97 to 2.37) | 0.07 |

*Adjusted for ethnicity, deprivation, school category and clustering by school.
aOR, adjusted OR.

2 (parental verbal consent and adolescent self-consent), there was no evidence for narrowing of this gap (p=0.27). However, after stage 3 (community catch-up clinics and family practice settings), the decrease in unvaccinated young women was greater for those that attended alternative education provider settings, compared with those who attended comprehensive, non-fee paying schools (accumulative risk difference: 23.7% vs 10.2%, p<0.001) (figure 1 and table 3).

There were also differences by ethnicity. At stage 1 (parental written consent only), 15.8% of white British young women were unvaccinated, in comparison to 26.7% of non-white British young women and 35.7% belonging to the unknown ethnic category. Following implementation of stage 2 (parental verbal consent and adolescent self-consent), the percentage unvaccinated decreased at a greater rate in young women classified as 'unknown' ethnic category compared with white British young women (risk difference: 6.7% vs 4.2%, p<0.001), but no difference was observed for non-white British young women (p=0.21). There was evidence for a difference with the inclusion of stage 3 for non-white British women (accumulative risk difference: 12.8% vs 8.9%, p=0.01) and unknown ethnicity (accumulative risk difference: 13.4% vs 8.9%, p<0.001) (figure 1 and table 3).

Inequalities in the percentage of unvaccinated young women by deprivation were attenuated by each stage of implementation of the policy. At stage 1 (parental written consent only), 14.2% young women in the least deprived quintile were unvaccinated in comparison to 29.8% of young women in the most deprived quintile. Subsequent to stage 2 (parental verbal consent and adolescent self-consent), the percentage of unvaccinated young women decreased at a greater rate for those belonging to the most deprived quintile compared with the least deprived quintile (risk difference: 7.4% vs 2.3%, p<0.001). A similar pattern was observed following implementation of stage 3 (community catch-up clinics and family practice settings) (overall risk difference: 13.8% vs 5.8%, p<0.001) (figure 1 and table 3).

## DISCUSSION

The HPV vaccination programme has been implemented to prevent acquisiton of HPV, a recognised precursor to developing cervical cancer. The findings from this study show that some of the barriers to young women being vaccinated were overcome through the implementation of a new local policy, which included parental verbal consent and adolescent self-consent in the school setting. There is promising evidence that the additional steps have the potential to reduce existing inequalities in uptake among young women living in more deprived areas. Importantly, this study showed that these young women are less likely to engage with consent procedures that rely on paper-based methods, are more likely to be affected by cervical cancer,[5 15] and less likely to receive the HPV vaccine.[6]

The majority of young women were vaccinated in the school setting. However, this study supports the

**Table 3** Percentage of young women unvaccinated by implementation stage of consent process

| | N (%) | Stage 1‡ n (%) | Stages 1 and 2 n (%) | Risk reduction* n (%) | P value* | Stages 1, 2 and 3 n (%) | Risk reduction† n (%) | P value† |
|---|---|---|---|---|---|---|---|---|
| | | 1519 (21.3) | 1173 (16.5) | 347 (4.8) | – | 788 (11.1) | 731 (10.3) | – |
| | N (%) | % (95% CI) | % (95% CI) | % (95% CI) | | % (95% CI) | % (95% CI) | |
| **School category** | | | | | | | | |
| Comprehensive, non-fee paying | 6350 | 20.6 (19.6 to 21.6) | 15.4 (14.5 to 16.3) | 5.3 (4.7 to 5.8) | – | 10.4 (9.7 to 11.2) | 10.2 (9.5 to 11.0) | – |
| Private, fee paying | 661 | 22.7 (19.7 to 26.0) | 21.5 (18.5 to 24.8) | 1.2 (0.6 to 2.4) | <0.001 | 14.5 (12.0 to 17.4) | 8.2 (6.3 to 10.5) | 0.07 |
| Alternative education providers | 118 | 50.8 (41.9 to 59.7) | 47.5 (38.7 to 56.4) | 3.4 (1.3 to 8.7) | 0.27 | 27.1 (19.9 to 35.8) | 23.7 (16.9 to 32.3) | <0.001 |
| **Ethnicity** | | | | | | | | |
| White British | 4705 | 15.8 (14.8 to 16.8) | 11.6 (10.7 to 12.5) | 4.2 (3.6 to 4.8) | – | 6.9 (6.2 to 7.6) | 8.9 (8.1 to 9.7) | – |
| Non-white British | 542 | 26.7 (23.3 to 30.5) | 21.3 (18.2 to 24.9) | 5.4 (3.8 to 7.6) | 0.21 | 14.0 (11.4 to 17.1) | 12.8 (10.3 to 15.8) | 0.01 |
| Unknown | 1503 | 35.7 (33.4 to 38.0) | 29.0 (26.9 to 31.2) | 6.7 (5.6 to 8.0) | <0.001 | 22.3 (20.4 to 24.3) | 13.4 (11.8 to 15.1) | <0.001 |
| **Deprivation** | | | | | | | | |
| Least deprived | 1279 | 14.2 (12.4 to 16.1) | 11.9 (10.3 to 13.7) | 2.3 (1.6 to 3.3) | – | 8.4 (7.0 to 10.0) | 5.8 (4.7 to 7.2) | – |
| Quintile 2 | 1324 | 14.9 (13.1 to 16.8) | 11.2 (9.7 to 13.0) | 3.6 (2.8 to 4.8) | 0.04 | 7.4 (6.1 to 8.9) | 7.5 (6.2 to 9.0) | 0.08 |
| Quintile 3 | 1334 | 19.8 (17.8 to 22.0) | 14.9 (13.1 to 16.9) | 4.9 (3.9 to 6.2) | <0.001 | 9.3 (7.9 to 10.9) | 10.5 (9.0 to 12.3) | <0.001 |
| Quintile 4 | 1305 | 27.1 (24.8 to 29.5) | 20.8 (18.8 to 23.1) | 6.2 (5.1 to 7.6) | <0.001 | 13.8 (12.1 to 15.7) | 13.3 (11.6 to 15.1) | <0.001 |
| Most deprived | 1347 | 29.8 (27.4 to 32.2) | 22.4 (20.3 to 24.6) | 7.4 (6.1 to 8.9) | <0.001 | 16.0 (14.2 to 18.0) | 13.8 (12.1 to 15.7) | <0.001 |
| Unknown | | 24.2 (18.5 to 30.9) | 22.0 (16.6 to 28.5) | 12.6 (8.6 to 18.2) | 0.93 | 12.6 (8.5 to 18.3) | 11.5 (7.6 to 17.1) | 0.02 |

*Risk difference for stages 1 and 2 compared with stage 1; p value for comparison with risk difference in baseline group.
†Risk difference for stages 1, 2 and 3 compared with stage 1; p value for comparison with risk difference in baseline group.
‡Stage 1: parental written consent only; stage 2: schools-based procedures (parental verbal consent and adolescent self-consent; stage 3: community settings (catch-up clinics and family practices).

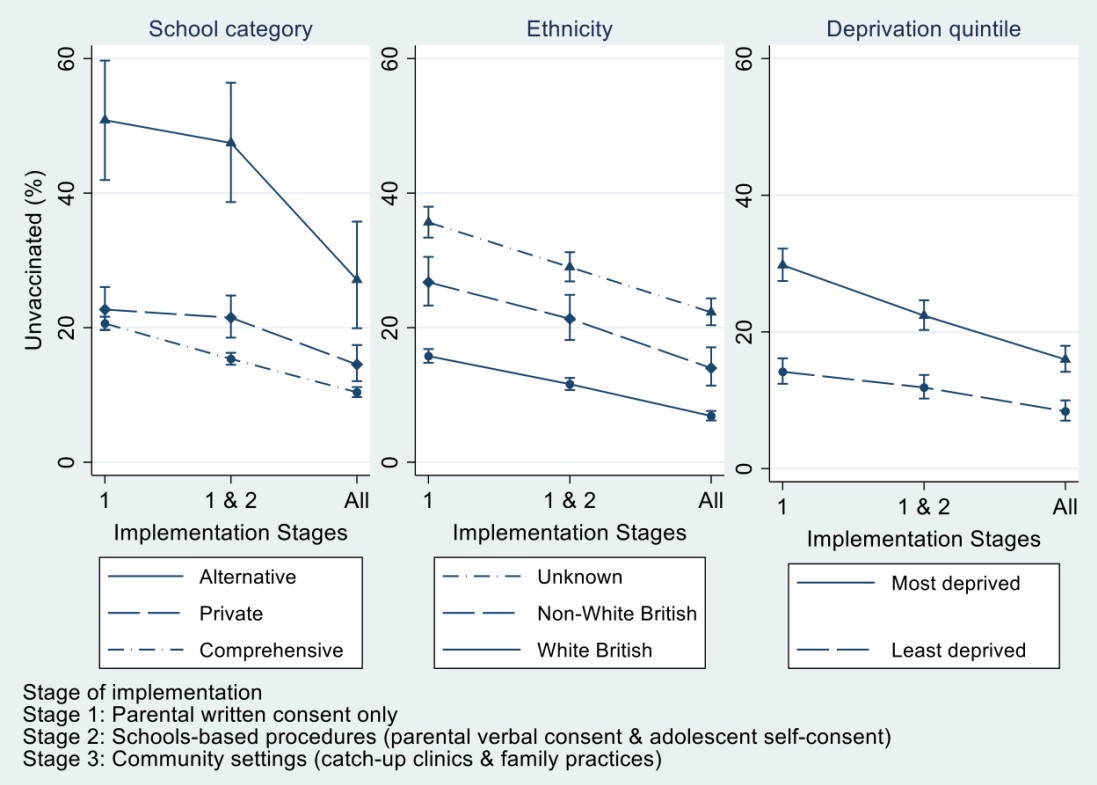

**Figure 1** Percentage of HPV vaccine eligible young women unvaccinated by stage of implementation of consent procedure. HPV, human papillomavirus.

provision of the HPV vaccine in community settings, such as catch-up clinics and general practice surgeries, to help improve access to vaccination. This may benefit young women who may have had anxieties about being vaccinated without a parent or carer present, or did not attend school on the day of the vaccination session.

Community provision of the HPV vaccine also appeared to reduce substantial inequalities in uptake among young women educated in alternative education provider settings. The reasons for this are unclear, but could relate to their lower school attendance, complex behavioural and physical health needs, or perceptions of safety of vaccination in the school setting due to interactions with other medical treatment. Although they comprise a small proportion of the overall vaccine-eligible population, they are a vulnerable groups with substantially lower uptake and greater health inequalities which requires addressing.

As the majority of parents who could be contacted provided verbal consent, the data imply that the absence of a signed parental consent form cannot be assumed to mean the parent does not want their daughter to have the vaccine. Not all parents could be contacted by the immunisation team on the day of the vaccination session. Provision of additional resources to contact families ahead of the vaccination session could help reduce the proportion of families who are not contacted and help ensure that their daughter receives the HPV vaccine if they wish.

Relative to parental verbal consent, adolescent self-consent occurred infrequently. Our analyses of qualitative data from this study showed a strong presumption that parents should make decisions affecting the health of their children. The preferred age at which the HPV vaccination is administered (12–13 years) also contributed to reluctance in endorsing self-consent which was thought to have the potential to break down trust between parents and school staff, and within families.[16] This suggests that unresolved issues could act as a barrier to widespread implementation of adolescent self-consent in other settings.

Our primary statistical analyses for this study showed that the new consent procedures increased uptake by 11% in one of the intervention local authorities, and appeared to overcome trends for decreasing uptake in matched sites.[11] Our secondary analyses of the process evaluation data reported here showed an additional 347 young women (4.8%) received the HPV vaccine in the school setting. All things being equal, if changes in policy resulted in similar effect sizes in other local authorities, as part of a strategy to increase uptake, then the English HPV vaccination programme could reach the WHO's target of 90% of young women receiving the vaccination by 15 years old.[17]

Additionally, establishing the cost-effectiveness of strategies to improve uptake of vaccination programmes is important to provide evidence for policy-makers to target

resources appropriately. This has been established in the context of the USA,[18] but these findings are not easily translatable to English schools-based, rather than health-carebased, vaccination programmes.

Public Health England have recently issued updated guidance for healthcare professionals related to the new universal HPV vaccination programme.[19] This supports the use of parent verbal consent and adolescent self-consent as strategies to maximise uptake and reduce catch-up sessions. They cite further benefit of inclusion of young people whose parents may have difficulties in completing the consent due to language or literacy issues. The findings from this study provide evidence that strategies incorporating parent verbal consent could help young women belonging to 'harder-to-reach' families receive the HPV vaccine. These recommendations may also be applicable to other schools-based vaccination programmes, including the influenza vaccination programme offered to primary school-aged children where similar patterns in forms returns have been reported.[20]

### Strengths and limitations

The study has some strengths. This is the first study to examine how new local polices for the HPV vaccination programme are implemented, and the impact on health inequalities among more deprived populations and minority ethnicity groups. Our study used routinely collected data related to vaccination status, eliminating the risk of recall and selection bias. The data relate to vaccinations delivered in school and community settings to all young women eligible for routine HPV vaccination during the study period. As such, our results correspond to an almost complete population.

There are some limitations. The data related to implementation of new consent procedures in a geographically distinct area in the south-west of England. The findings therefore may not be applicable to local authorities that are implementing new consent procedures in schools-based vaccination programmes elsewhere in the UK. The findings may also not translate to other adolescent vaccination programmes delivered in countries where cultural differences may influence the acceptability of parental verbal consent and adolescent self-consent procedures in the school setting.

As the study relied on routinely collected information, we did not have access to individual-level measures of socioeconomic status and relied on area-based measures of deprivation. Our study findings may therefore be subject to ecological fallacy.

An issue, common to all routinely collected data, is the possibility of data input errors and missing data. To minimise bias from inclusion of this data, we excluded almost 5% of the data as the information related to school was out-of-date. Overall, our dataset identified 8% of young women belonging to a minority ethnic group. This compares with nationally reported figures indicating) 30% of young people attending secondary schools in the intervention areas belong to a non-white ethnic group.[21]

Missing ethnicity data (23%) relating to young women who were born outside the local authority boundaries could change the direction or size of aORs corresponding to ethnicity.

## CONCLUSIONS

Introducing further steps to the consent procedures—allowing parents to consent verbally and adolescent self-consent—overcame some of the barriers to vaccination of young women belonging to families less likely to respond to paper-based methods of gaining consent.

**Author affiliations**
[1]Bristol Medical School, University of Bristol, Bristol, UK
[2]Sirona Care and Health CIC, Kingswood, UK
[3]Screening and Immunisations South West, Public Health England, London, UK
[4]Department of Paediatric Immunology & Infectious Diseases, University Hospitals Bristol and Weston NHS Foundation Trust, Bristol, UK
[5]Bristol Biomedical Research Centre and NIHR CLAHRC West, University of Bristol, Bristol, UK

**Acknowledgements** The authors express their gratitude to staff at In Health Intelligence for data linkage and extracting the data from the Child Health Information System, and to the immunisation and administrative teams at Sirona Healthcare who facilitated access to the immunisation consent records as part of this study. The study is partly supported by the NIHR Health Protection Research Unit in Evaluation of Interventions at University of Bristol. This research was supported by the National Institute for Health Research (NIHR) Applied Research Collaboration West (NIHR ARC West). The work was undertaken with the support of The Centre for the Development and Evaluation of Complex Interventions for Public Health Improvement (DECIPHer), a UKCRC Public Health Research Centre of Excellence. Joint funding (MR/KO232331/1) from the British Heart Foundation, Cancer Research UK, Economic and Social Research Council, Medical Research Council, the Welsh Government and the Wellcome Trust, under the auspices of the UK Clinical Research Collaboration, is gratefully acknowledged.

**Contributors** SA, HF, MH, KE, JF and MR were involved in the conception and design of the research. SA is principal investigator; HF is study manager and undertook data cleaning, analysed the data and drafted the manuscript; JF, JY and KE developed and advised on the new consent procedures and the HPV vaccination process; MR and KE advised on local immunisation strategy; JM advised on health inequalities and MH and RR advised on statistical methods. All authors have made substantial contributions to interpreting the data, revising it for important intellectual content and have given approval of the final version to be submitted.

**Funding** This work is funded by the National Institute for Health Research's Research for Patient Benefit (NIHR RfPB) programme (project number PB-PG-0416–20013).

**Disclaimer** The views expressed in this article are those of the author(s) and not necessarily those of the NIHR or the Department of Health and Social Care.

**Competing interests** None declared.

**Patient consent for publication** Not required.

**Provenance and peer review** Not commissioned; externally peer reviewed.

**Data availability statement** Data may be obtained from a third party and are not publicly available. As the analysis was undertaken on routine data acquired from an external source, the study authors assured the data custodians that would the dataset would be treated as confidential and would not be shared.

**ORCID iDs**
Harriet Fisher http://orcid.org/0000-0002-5639-0955
Matthew Hickman http://orcid.org/0000-0001-9864-459X

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
