## [Reviewer comments · BMJ Open]

ARTICLE DETAILS

TITLE (PROVISIONAL)	Secondary analyses to test the impact on inequalities and uptake of the schools-based human papillomavirus (HPV) vaccination programme by stage of implementation of a new consent policy in the south-west of England
AUTHORS	Fisher, Harriet; Evans, Karen; Reynolds, Rosy; Yates, Julie; Roderick, Marion; Ferrie, Jo; Macleod, John; Hickman, Matthew; Audrey, Suzanne

VERSION 1 – REVIEW

REVIEWER	ME Cruickshank University of Aberdeen, Aberdeen Reserach Centre for Women's Health
REVIEW RETURNED	18-Dec-2020

GENERAL COMMENTS	Uptake of vaccination is fundamental to the success of the programme and this paper looks at uptake in relation to consecutive measures to improve consent and therefore up take in relation to deprivation, ethnicity and education setting. Effective use is made of routinely collected data to describe the outcomes and the limitations of this approach and ability to define these groups is given. The results provide an opportunity to consider and debate such health inequalities and differences in uptake. Although the use of routinely collected data allows for a large cohort on population basis, the results generate more questions than answers - which is not necessarily a problem. It would help to have some more context for example, in Scotland, parental/guardian discussion and consent is advised but not required so are there difference between the 4 nations in the UK? How does provision of consent and uptake compare with childhood immunisations compared with adolescent uptake here? The use of community clinics, which are not usually considered to be as effective as school based programmes, appear to be used more by girls in alternative education providers and I would be interested to know why this might be. This paper appears to be a service evaluation and as I clinician, I am still wondering what does it mean to me. I was not clear if the difference in uptake by intervention was of clinical significance at a population level or what evidence was used to decide on these interventions. Indeed, was there added benefit in getting verbal consent from parents compared with assessing competency and getting consent from the girl herself. By the end of the discussion, i would have liked to have more evidence to support introduction of interventions. alternatively, if such differences in uptake, given high uptake overall, can translate into reductions in cervical abnormalities, what are next steps to evaluate these and identify what may be more clinically and cost effective.
---

REVIEWER	Albert Lee The Chinese University of Hong Kong, School of Public Health and Primary Care
REVIEW RETURNED	20-Jan-2021

GENERAL COMMENTS	This study addresses important issues of public health concern. The study investigates the factors associated with parental consent and effect of new policy on consent and also stages of implementation. . Univariate logistic regression was utilised to analyse factors associated with parental consent. It was stated that multivariable logistic regression model was developed. However, no results are presented to show the independent factors associated with consent. The table in supplementary material provides important findings. It should be presented in main text. Some factors such as unknown ethnicity, alternate education providers, certain quintiles of deprivation show decreasing trend of unvaccinated proportion. One would consider statistical analysis of the trend for significance. Stage 3 implementation appears to have significant impact. Engagement of catch up clinics and family practice would make a difference. The discussion section should strengthen on this perspective. This findings highlight the importance of community based care and role of family doctors/general practitioners in providing comprehensive and holistic care particularly preventive medicine. The findings of this study support better compliance if care is delivered in community particularly by their own family doctors. There should be more discussions drawing on latest literature.
--

REVIEWER	Linda Selvey University of Queensland, Faculty of Medicine
REVIEW RETURNED	31-Jan-2021

GENERAL COMMENTS	This is an important evaluation of an initiative to increase HPV vaccination consent and uptake. However the methods are unclear. The authors state that this was a cross sectional study yet the study compares outcomes at three different time points. How this is done is not apparent. Was the full cohort exposed to all three interventions or was it a subset of the cohort that were exposed to stage 1; another stage 2; etc. The methods do not state this clearly. If it is the same group exposed to all three stages then a simple comparison between the three stages is not appropriate. Was it just that the students had three different opportunities to be vaccinated? As the study went over two years it is not clear how this might have happened. If subsets of the cohort received either stages 1, 2 or 3, then the number in each group should be provided, including the demographics of each group. This manuscript needs to be re-written in order to properly assess what was done. Secondly a risk difference measure was used to describe differences between coverage in each phase. Why was this measure used? Usually a risk ratio would be the measure. This need to be explained. Finally, some of the wordings needs to be looked at. For example, you wouldn't normally say 'strong evidence of' in the results section. Results should just be shown.
--

VERSION 1 – AUTHOR RESPONSE

We thank the reviewers for their insightful comments and to the editor for providing us with the opportunity to respond. We have addressed each point in turn below.

Reviewer: 1 Uptake of vaccination is fundamental to the success of the programme and this paper looks at uptake in relation to consecutive measures to improve consent and therefore up take in relation to deprivation, ethnicity and education setting. Effective use is made of routinely collected data to describe the outcomes and the limitations of this approach and ability to define these groups is given. The results provide an opportunity to consider and debate such health inequalities and differences in uptake. Although the use of routinely collected data allows for a large cohort on population basis, the results generate more questions than answers - which is not necessarily a problem. It would help to have some more context for example, in Scotland, parental/guardian discussion and consent is advised but not required so are there difference between the 4 nations in the UK? How does provision of consent and uptake compare with childhood immunisations compared with adolescent uptake here?	To our knowledge, there are no differences within the legal framework between the four nations within the UK. However, there is evidence for differences in interpretation of how the law should be applied at the local level. Although we are aware of research that has examined this issue within the English and Welsh context, we are unable to confirm to what extent this occurs specifically within the Scottish context. This paper focuses on an English adolescent vaccination programme which is delivered in a different setting to childhood immunisation programmes. Factors affecting uptake are complex, and not solely related to consent for either vaccination programme. For these reasons, we would prefer not to draw comparisons with childhood and adolescent vaccination programmes as advised. Page 4, paragraph 3. However we agree with the reviewer that clarification would be helpful. We have provided the reader with an overview of the consent procedures for adolescent vaccination programmes across the four nations, prior to setting out the legal framework: 'In the United Kingdom (UK) (comprising England, Scotland, Northern Ireland, and Wales), consent for schools-based adolescent vaccination programmes is usually obtained from parents or carers. Young people are provided with a form for their parent to sign and hand back to school before a vaccination session takes place.'
The use of community clinics, which are not usually considered to be as effective as school based programmes, appear to be used more by girls in alternative education providers and I would be interested to know why this might be.	This is an interesting and important point. Unfortunately, in this paper we can only hypothesise the reasons as we lack data to determine the reason. Page 12, paragraph 3. We have alluded to

	potential reasons for greater use of community-settings by young women attending alternative education settings as follows: ‘Community provision of the HPV vaccine also appeared to reduce substantial inequalities in uptake among young women educated in alternative education provider settings. The reasons for this are unclear, but could relate to their lower school attendance, complex behavioural and physical health needs, or perceptions of safety of vaccination in the school setting due to interactions with other medical treatment. Although they comprise a small proportion of the overall vaccine-eligible population, they are a vulnerable population with substantially lower uptake and greater health inequalities which requires addressing.’
This paper appears to be a service evaluation and as I clinician, I am still wondering what does it mean to me.	Abstract & page 12, paragraph 1. We apologise that this is not clear. We have amended statements within the abstract and discussion to emphasise that the introduction of the new consent policy overcomes some of the barriers to uptake and could help reduce health inequalities related to this immunisation programme which has been implemented to prevent HPV, a recognised precursor to developing cervical cancer.
I was not clear if the difference in uptake by intervention was of clinical significance at a population level or what evidence was used to decide on these interventions. Indeed, was there added benefit in getting verbal consent from parents compared with assessing competency and getting consent from the girl herself.	Page 13, paragraphs 3 & 4. We have updated the discussion to include reference to our recently published paper which reports the overall impact of the consent procedures on uptake of the HPV vaccination programme. We are unsure of thresholds for clinical significance with regards to the HPV vaccination programme. However, we have also made reference to the World Health Organisation’s strategy to eliminate cervical cancer which includes a target of 90% of young women being vaccinated by 15 years old. Page 13, paragraph 2. We discuss how adolescent self-consent occurred infrequently compared to parental verbal consent, in addition to reporting barriers to implementing adolescent

	self-consent procedures.
By the end of the discussion, i would have liked to have more evidence to support introduction of interventions. alternatively, if such differences in uptake, given high uptake overall, can translate into reductions in cervical abnormalities, what are next steps to evaluate these and identify what may be more clinically and cost effective.	Page 13, paragraphs 3 & 4. We are unsure of vaccination uptake thresholds which translate to clinical reductions in cervical abnormalities. However, to address the reviewer’s comment we have included the following paragraph which makes reference to the World Health Organisation’s strategy to eliminate cervical cancer and a study examining cost-effectiveness of interventions to improve uptake. ‘All things being equal, if changes in policy resulted in similar effect sizes in other local authorities as part of a strategy to increase uptake, then the English HPV vaccination programme could reach the World Health Organisation’s target of 90% of young women receiving the vaccination by 15 years old [18]. Additionally, establishing the cost-effectiveness of strategies to improve uptake of vaccination programmes is important to provide evidence for policy makers to target resources appropriately. This has been established in the context of the United States of America [19], but these findings are not easily translatable to English schools-based, rather than healthcare-based, vaccination programmes.’
Reviewer: 2 This study addresses important issues of public health concern. The study investigates the factors associated with parental consent and effect of new policy on consent and also stages of implementation. Univariate logistic regression was utilised to analyse factors associated with parental consent. It was stated that multivariable logistic regression model was developed. However, no results are presented to show the independent factors associated with consent.	Page 10, paragraph 2. We have presented information pertaining to the independent variables (education setting, ethnic group & deprivation quintile) as adjusted Odds Ratios within the body of the manuscript. We have clarified that the unadjusted Odds Ratios are provided in Table 2.
The table in supplementary material provides important findings. It should be presented in main text. Some factors such as unknown ethnicity, alternate education providers, certain quintiles of deprivation show decreasing trend of unvaccinated proportion. One would consider statistical analysis of the trend for significance.	We have now presented the information within Table 3, instead of as supplementary material, as suggested. We are unclear whether the reviewer is referring to trends in uptake or risk reduction. If the reviewer is referring to uptake, we know

	disparities in vaccination uptake according to such factors are well established, and the study did not set out to identify them. The purpose was to investigate by how much changing the consent process could increase uptake and so potentially reduce disparity. If the reviewer is referring to risk reduction, our justification for response is the following: As each 'stage' of implementation of the consent process is completed and young women are offered additional opportunities to be vaccinated, we know that the proportion of unvaccinated young women can only go down (or stay the same) as each stage is completed, because only unvaccinated young women enter each successive stage. The table already shows p-values comparing the risk reduction with that in the baseline group. However, it would not be appropriate to look for trends across categories of variables (ethnicity, school deprivation, quintile) because they have no inherent ordering.
Stage 3 implementation appears to have significant impact. Engagement of catch up clinics and family practice would make a difference. The discussion section should strengthen on this perspective. This findings highlight the importance of community based care and role of family doctors/general practitioners in providing comprehensive and holistic care particularly preventive medicine. The findings of this study support better compliance if care is delivered in community particularly by their own family doctors. There should be more discussions drawing on latest literature.	Page 12, paragraph 2. The majority of young women were successfully vaccinated in the school setting, with a relatively lower proportion of the population receiving the HPV vaccine through community-based settings. However, we agree with the reviewer's perspective in relation to the importance of provision of the HPV vaccine in community-based settings. Within the discussion section, we have highlighted under what circumstances this might be helpful through the following: 'The majority of young women were vaccinated in the school setting. However, this study supports the additional provision of the HPV vaccine in community settings, such as catch-up clinics and general practice surgeries, to help

	improve access to vaccination. Young women who may benefit from this include young women who may have had anxieties about being vaccinated without a parent present or did not attend school on the day of the vaccination session.'
Reviewer: 3 This is an important evaluation of an initiative to increase HPV vaccination consent and uptake. However the methods are unclear. The authors state that this was a cross sectional study yet the study compares outcomes at three different time points. How this is done is not apparent. Was the full cohort exposed to all three interventions or was it a subset of the cohort that were exposed to stage 1; another stage 2; etc. The methods do not state this clearly. If it is the same group exposed to all three stages then a simple comparison between the three stages is not appropriate. Was it just that the students had three different opportunities to be vaccinated? As the study went over two years it is not clear how this might have happened. If subsets of the cohort received either stages 1, 2 or 3, then the number in each group should be provided, including the demographics of each group. This manuscript needs to be re-written in order to properly assess what was done.	We apologise for the lack of clarity regarding the methodological approach for this study. We confirm that the study comprises a cross-sectional study design. We also confirm that each 'stage of consent' represents the point at which the vaccination was received, rather than a time-dependent variable. Sub-sets of the cohort were not 'exposed' to different stages as the reviewer suggests. Young women would only be 'exposed' to all three stages if they remain unvaccinated. For example, if the receipt of a written consent form completed by parents is confirmed, the young person would not be 'exposed' to further stages of the consent process, such as by the immunisation team seeking parental verbal consent or assessing the young person to self-consent. We have made the following amendments within the manuscript to clarify the points raised by the reviewer. Page 6, paragraph 2. Here we clarify the definition of stages of implementation of the consent procedures and that these represent opportunities for unvaccinated young women to receive the vaccination as follows: 'We define stage of implementation of consent procedures as the following: (i) 'Stage One: parent written consent only'; (ii) 'Stage Two: parent verbal consent & adolescent self-consent', and; (iii) 'Stage Three: community catch up clinics and family practice settings' These stages represent sequential opportunities (in a single programme) for unvaccinated young women to receive the

	vaccine, rather than different time-points.' Page 7, paragraph 2. We confirm the analyses do not span different time points. Rather, the data period of the study spanned the HPV vaccination programme years 2017-2018 and 2018/2019. Page 8, paragraph 3. We have detailed the classification of each record to the stages of consent as follows: 'We classified each record as belonging to one of the following stages of the consent procedure: (i) 'Stage One: parent written consent only'; (ii) 'Stage Two: parent verbal consent & adolescent self-consent', and; (iii) 'Stage Three: community catch up clinics and family practice settings'. Records could not be assigned to more than one stage of consent category.'
Secondly a risk difference measure was used to describe differences between coverage in each phase. Why was this measure used? Usually a risk ratio would be the measure. This need to be explained.	Page 9, paragraph 2. We calculated risk differences (difference in two proportions), rather than risk ratios to describe the decrease in unvaccinated young women at each stage of implementation of the consent procedures. We have clarified within the manuscript that this shows the absolute effect of implementation of each stage of the consent procedure.
Finally, some of the wording needs to be looked at. For example, you wouldn't normally say 'strong evidence of' in the results section. Results should just be shown.	Page 10, paragraph 2. We have removed the reference to 'strong evidence of' and reworded the sentence as follows: 'After adjusting for school category, ethnicity, and deprivation, an association was found between parental consent form not being returned and: attending an alternative education provider setting (adjusted Odds Ratio (aOR): 5.54, 95% CI: 3.80-8.09, p<0.001); belonging to a Non-White British (aOR: 1.34, 95% CI: 1.06-1.70, p<0.01), and; Unknown ethnicity category (aOR: 2.41, 95% CI: 2.09-2.78, p<0.001).'